# A Nonlinear Radiometric Normalization Model for Satellite Imgaes Time Series Based on Artificial Neural Networks and Greedy Algroithm

Zhaohui Yin, Lejun Zou, Jiayu Sun, Haoran Zhang, Wenyi Zhang and Xiaohua Shen *

Key Laboratory of Geoscience Big Data and Deep Resource of Zhejiang Province, School of Earth Sciences, Zhejiang University, Hangzhou 310027, China; zhyin@zju.edu.cn (Z.Y.); zoulejun2006@zju.edu.cn (L.Z.); sunjy2016@zju.edu.cn (J.S.); zhanghr5@mail2.sysu.edu.cn (H.Z.); 3170101586@zju.edu.cn (W.Z.)
* Correspondence: shenxh@zju.edu.cn (X.S.)

**Abstract:** Satellite Image Time Series (SITS) is a data set that includes satellite images across several years with a high acquisition rate. Radiometric normalization is a fundamental and important preprocessing method for remote sensing applications using SITS due to the radiometric distortion caused by noise between images. Normalizing the subject image based on the reference image is a general strategy when using traditional radiometric normalization methods to normalize multitemporal imagery (usually two or three scenes in different time phases). However, these methods are unsuitable for calibrating SITS because they cannot minimize the radiometric distortion between any pair of images in SITS. The existing relative radiometric normalization methods for SITS are based on linear assumptions, which cannot effectively reduce nonlinear radiometric distortion caused by continuously changing noise in SITS. To overcome this problem and obtain a more accurate SITS, we propose a nonlinear radiometric normalization model (NMAG) for SITS based on Artificial Neural Networks (ANN) and Greedy Algorithm (GA). In this method, GA is used to determine the correction order of SITS and calculate the error between the image to be corrected and normalized images, which avoids the selection of a single reference image. ANN is used to obtain the optimal solution of error function, which minimizes the radiometric distortion between different images in SITS. The SITS composed of 21 Landsat-8 images in Tianjin, China, from October 2017 to January 2019 was selected to test the method. We compared NMAG with other two contrast methods (Contrast Method 1 (CM1) and Contrast Method 2 (CM2)), and found that the average root mean square error ($\mu_{RMSE}$) of NMAG (497.22) is significantly smaller than those of CM1 (641.39) and CM2 (543.47), and the accuracy of normalized SITS obtained using NMAG increases by 22.4% and 8.5% compared with CM1 and CM2, respectively. These experimental results confirm the effectiveness of NMAG in reducing radiometric distortion caused by continuously changing noise between images in SITS.

**Keywords:** satellite image time series; radiometric normalization; nonlinear radiometric distortion; artificial neural networks

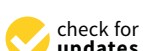

## 1. Introduction

Satellite Image Time Series (SITS) is a data set that includes satellite images across several years with a high acquisition rate. SITS can provide abundant information to describe temporal changes in the generation and development of ground features in an area [1]. Thus, it has been used as an important data source in many fields such as environmental monitoring, land cover change monitoring, crop growth monitoring, and so on [2–5]. However, the temporal information extracted from SITS is inevitably disturbed by noise unrelated to ground features, such as atmospheric absorption and scattering, sensor-target illumination geometry, sensor calibration, etc., which lead to inaccurate results in remote sensing applications [6]. Thus, radiometric normalization is required prior to any applications using SITS.

Radiometric calibration can be classified into absolute radiometric calibration and relative radiometric calibration (also named relative normalization). Because it is usually difficult to obtain atmospheric properties, performing relative radiometric calibration based on the inherent radiometric information of images provides an alternative when absolute surface radiances are not needed [7]. In early research methods, global image statistics were used to directly establish the gray value mapping relationship between the subject image and the reference image, such as Histogram Matching (HM), Mean-Standard deviation (MS), and so on [8,9]. However, these methods reduce the radiometric difference resulting from the variation in ground features while reducing the radiometric distortion caused by noise. Therefore, these methods are only suitable for image mosaicking due to the ambiguous result of the physical meaning of normalized images [10]. To effectively address this issue, methods based on the regression model have been developed, which is established using a set of invariant pixels [11] (named Pseudo-Invariant Features (PIFs)) from the subject and reference images.

Linear regression models are simple and effective, so they have been widely used to minimize radiometric distortion. Jenson et al. proposed a method based on a simple linear regression (SR) model, which obtains the linear relation between two scenes using the least square method [12]. Du et al. proposed an objective normalization procedure to find the most likely linear relation between the subject image and reference image through a principal component analysis (PCA) [13]. Olthof et al. developed a Theil-Sen regression model to avoid the error propagation caused by outliers [14]. Ghanbari et al. obtained the linear function between subject and reference images through the Error Ellipse (EE) process with a determined confidence level, which can removes the outliers from the linear model fitting [15]. However, relative normalization methods based on linear assumptions are inappropriate for dealing with complex nonlinear radiometric distortions between the subject image and the reference image. Radiometric normalization models based on artificial intelligence method, such as genetic algorithms [16], artificial neural network [17], random forest [18], kernel canonical correlation analysis [19], etc., have been used as an effective tool to deal with this problem.

The relative radiometric methods mentioned above are unsuitable for calibrating SITS, because they are based on the reference image to implement the radiometric normalization of the subject image, which cannot minimize the radiometric distortion between any pair of images in SITS. Wu et al. developed a new radiometric normalization procedure for SITS to effectively solve this issues, obtaining more objective and accurate correction results than previous methods [20]. However, continuously changing noise in SITS usually results in nonlinear radiometric distortion between images in reality, and the method proposed by Wu et al. cannot effectively reduce such nonlinear radiometric distortion.

An optimum method for minimizing nonlinear radiometric distortion between images in SITS is needed to facilitate remote sensing application. Thus, in this study we constructed a nonlinear radiometric normalization model for SITS, in which an artificial neural network (ANN) and greedy algorithm (GA) are combined for the nonlinear radiometric normalization of SITS. Here, GA is used for determining the correction order of SITS and calculating the error function of the image to be corrected, and ANN is used for obtaining the optimal solution of the error function. Hereafter, this nonlinear radiometric normalization procedure is called NMAG model.

## 2. Description of Methodology

The existing radiometric normalization approaches are based on the reference image $X_r$ to correct the subject image $X_c$. The key of these methods is finding a mapping function $f()$ to minimize the radiometric distortion between the subject image $f(X_c)$ and reference image $X_r$. The objective function $Q_r$ can be expressed as:

$$Q_r = min \sum_{s \in S} [f(X_c^s) - X_r^s]^2 \tag{1}$$

where $S$ represents Pseudo-Invariant Features (PIFs), $s \in S$ represents the Pseudo-Invariant Feature (PIF) in the PIFs, $f(X_c^s)$ represents the Digital Number (DN) value of PIF in the normalized subject image $f(X_c)$ , and $X_r^s$ represents the DN value of the PIF in the reference image.

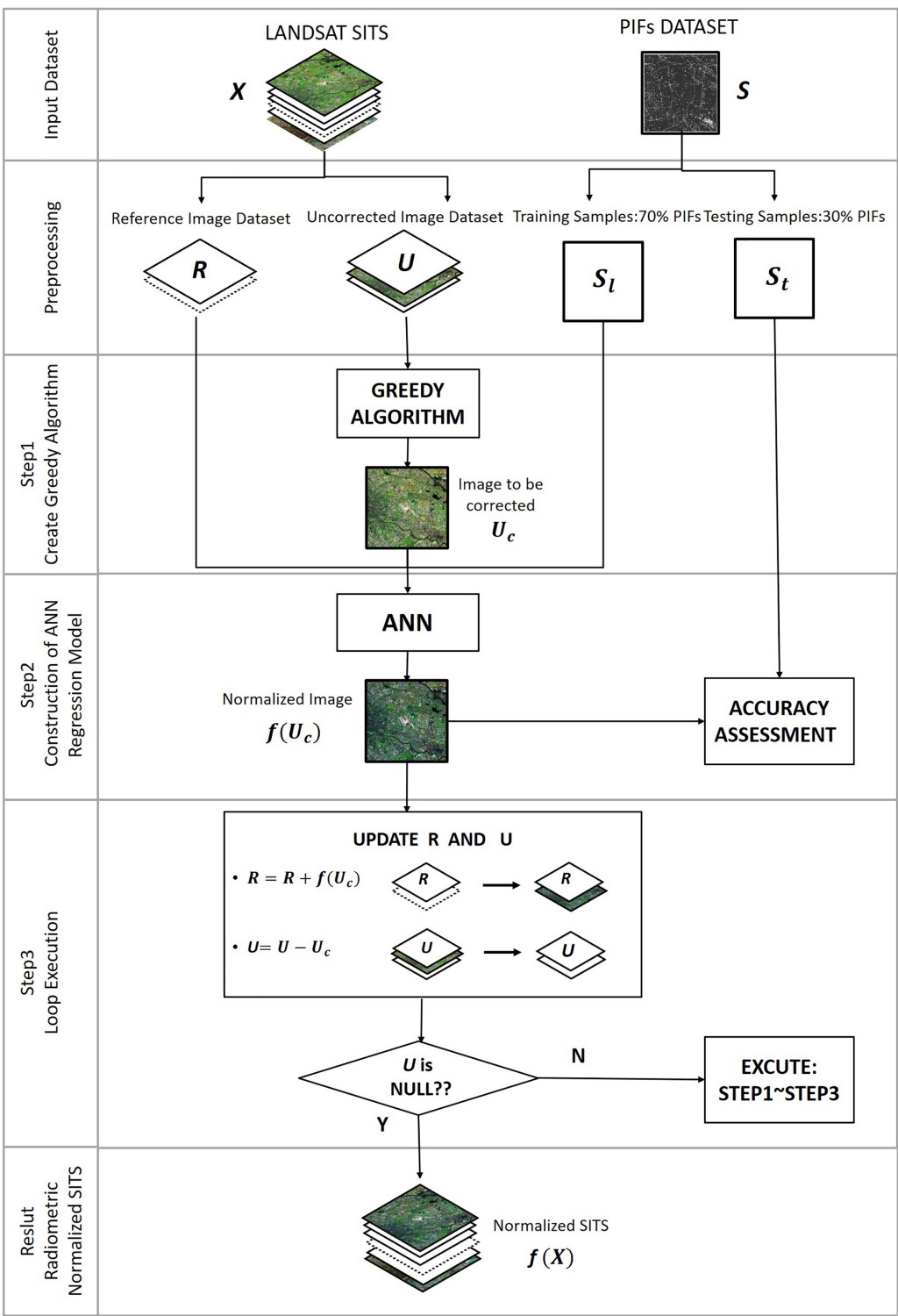

**Figure 1.** Flow diagram showing steps used by the proposed nonlinear radiometric normalization (NMAG) model for rediometric normalization of Satellite Image Time Series (SITS).

When Equation (1) is used for the correction of SITS $X$, an image $X_r \in X$ is selected as the reference image, and then other images $X_i$ in $X$ are normalized to the reference image $X_r$ one by one. Thus, the objective function $Q_r$ here can be written as:

$$Q_r = min \sum_{i=1}^{n} \sum_{s \in S} [f(X_i^s) - X_r^s]^2 \qquad (2)$$

where $n$ represents the number of images in $X$, and $f(X_i^s)$ represents the DN value of the PIF in the image to be corrected $f(X_i)$. It can be easily found from Equation (2) that $Q_r$ cannot ensure that the radiometric distortion between each pair of images in normalized SITS is minimum. Therefore, Wu et al. proposed an improved objective function $Q_G$ for the relative radiometric normalization of $X$, and $Q_G$ is expressed as [20]:

$$Q_G = min \sum_{i=1}^{n} \sum_{j=1}^{n} \sum_{s \in S} [f(X_i^s) - f(X_j^s)]^2 \qquad (3)$$

Our NMAG method is based on this objective function to achieve nonlinear radiometric normalization for SITS data by using artificial neural networks and greedy algorithms. The NMAG approach can be implemented using the following steps as shown in Figure 1, and the code of entire experiments are available from the public repositories (named NormSITS) (https://github.com/zhaohyin/NormSITS (accessed on 26 February 2021)) in GitHub.

Step 1. Create a greedy algorithm [21]. A greedy algorithm is an intuitive, well-tested algorithm used in optimization problems [22,23]. The algorithm makes the optimal choice at each step as it attempts to find the overall optimal method to solve the entire problem. To ensure that $Q_G$ can obtain the optimal solution, the greedy algorithm should be created to adopt the most greedy solution when implementing the rediometric normalization of each image in SITS.

SITS $X$ is divided into two groups, one as reference image dataset $R$ and the other as image dataset $U$ to be corrected:

$$X = R + U \qquad (4)$$

When NMAG has not yet started normalization, $R$ and $U$ can be expressed as:

$$R = \{ \ \} \ and \ U = X \qquad (5)$$

A clear and cloudless image $X_r$ is selected from SITS as the reference image, and then $X_r$ is added to the reference image dataset $R$. Update $U$ according to Equation (6), then an image with the smallest rediometric distortion from $R$ is selected from $U$ as the next image to be corrected $U_c$. The selection of $U_c$ can be given by Equation (7).

$$R = \{X_r\} \ and \ U = X - X_r \qquad (6)$$

$$c = \underset{x \in [1,m]}{argmin}(\sum_{i=1}^{k} \sum_{s \in S} (U_x^s - R_i^s)^2) \qquad (7)$$

where $k$ represents the number of normalized images in $R$, $m$ represents the number of uncorrected images in $U$, and $s \in S$ represents the PIF in PIFs. $R_i^s$ represents the DN value of PIF in the $i$-th image of $R$, and $U_x^s$ represents the DN value of the PIF in the $x$-th image in $U$.

Next, implement the radiometric normalization of $U_c$, and the local optimal solution $Q_c$ can be calculated by:

$$Q_c = min \sum_{i=1}^{k} \sum_{s \in S} [f(U_c^s) - R_i^s]^2 \qquad (8)$$

where $f(U_c)$ represents the radiometric normalization result for $U_c$, and $f(U_c^s)$ represents the DN value of the PIF in $f(U_c)$.

Step 2. Generate an ANN regression model. The linearity assumption of radiometric distortion in SITS is imprecise, and the nonlinear regression model may achieve more accurate normalization results of SITS. As a representative method of machine learning, ANN regression model has been widely used to solve nonlinear regression problems in remote sensing applications and has achieved significant results [24,25]. The schematic diagram of ANN [26] is shown in Figure 2.

**Figure 2.** The schematic diagram of the artificial neural network (ANN) regression model used in NMAG.

In this study, PIFs were randomly divided into a training sample set $S_l$ and a test sample set $S_t$, accounting for 70% and 30% of total PIFs respectively. $S_l$ was used for the training of ANN regression model, and $S_t$ was used for the accuracy assessment of ANN regression model after training. As shown in Figure 2, ANN includes an input layer, an output layer, and several hidden layers. The input layer is the DN value of the training samples in the image $U_c$ to be corrected, which is expressed as $U_c^s, s \in S_l$. The output layer is the radiometric normalization results $f(U_c^s), s \in S_l$. The experimental details of training and testing ANN, including the number of training samples and testing samples, the parameter of ANN, the ANN error convergence graphs for training etc., are recorded in detail in the Technical Document. This document is also available from the the public repositories (named NormSITS) (https://github.com/zhaohyin/NormSITS/blob/master/README.md (accessed on 26 February 2021)) in GitHub.

The ANN regression model, which consisted of an input–output pair, can be adjusted by the connection weights between the nodes by learning to memorize every one of the network learning and training samples. The magnitude of weights represents the importance of the link between the neurons to $f(U_c^s)$. The value of weights is initialized randomly, thus, the initial output is also random. The difference (i.e., error) between the generated output and a training set output is calculated and is fed back to the network, where it is used for connection-weight readjustment by Gradient Descent Algorithm (GDA) to minimize the error to within a predefined tolerance. In this way, the iteration calculation can reduce the error to a predetermined allowable range. The error estimation function can be expressed as:

$$LOSS = \frac{1}{2} * \sum_{i=1}^{k} \sum_{s \in S_l} (f(U_c^s) - R_i^s)^2 \tag{9}$$

where $k$ represents the number of images in $R$, and $R_i^s$ represents the DN value of PIFs in the $i$-th image in $R$.

Step 3. Loop execution. Add the corrected image $f(U_c)$ to reference image dataset $R$ and remove $U_c$ from uncorrected image dataset $U$. The update of $R$ and $U$ can be expressed as:

$$R = R + f(U_c) \; and \; U = U - U_c \tag{10}$$

where $f(U_c)$ is the radiometric normalization results of image $U_c$.

If images still exist in the updated $U$, we execute the relative radiometric correction of the next image in the same way.

## 3. Materials

### 3.1. Study Area and Satellite Data

Here, to illustrate the application of NMAG modle, SITS with 21 high-quality Landsat-8 satellite images covering an area of Tianjin, China, were used in this study. The acquisition time of these images ranges from October 2017 to January 2019 and the average cloud cover of images is 3.58%. The cost time of performing relative radiometric correction on entire images in SITS is too long; thus, only 1000*1000 pixels were selected as the study area. The false color composite image of the study area is shown in Figure 3.

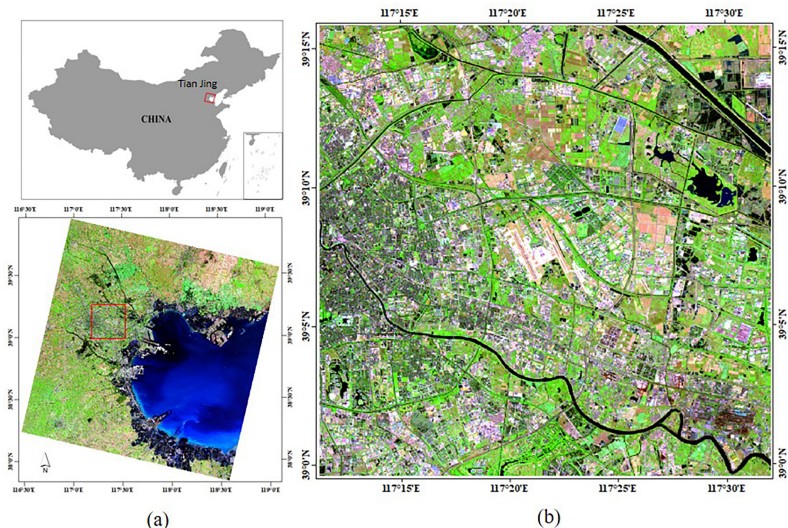

|       |       |
| :---: | :---: |
| (a)   | (b)   |

**Figure 3.** Overview of the study area: (**a**) the Landsat-8 image with Path=122 and Row=38 in the Worldwide Reference System (WRS); (**b**) The false color composite image in the study area (R: B7, G: B5, B: B4).

No further geometric correction is needed during image preprocessing because the positioning accuracy of these images obtained from the United States Geological Survey (USGS) (https://glovis.usgs.gov (accessed on 26 February 2021)) is less than one pixel. The false-color composite images of each temporal datum in the SITS is given in Figure 4, and we can find from this figure that this area is rich in ground features, of which artificial buildings are mainly concentrated in the urban area of Tianjin, bare land is mainly distributed in the suburbs of Tianjin, and road pixels are evenly distributed in the study area. The PIFs were mainly selected from pixels corresponding to these three types of ground features. Vegetation with intra-year seasonal changes are mainly distributed in suburbs of Tianjin, which can be selected to assess the accuracy of the radiometric normalization results of SITS.

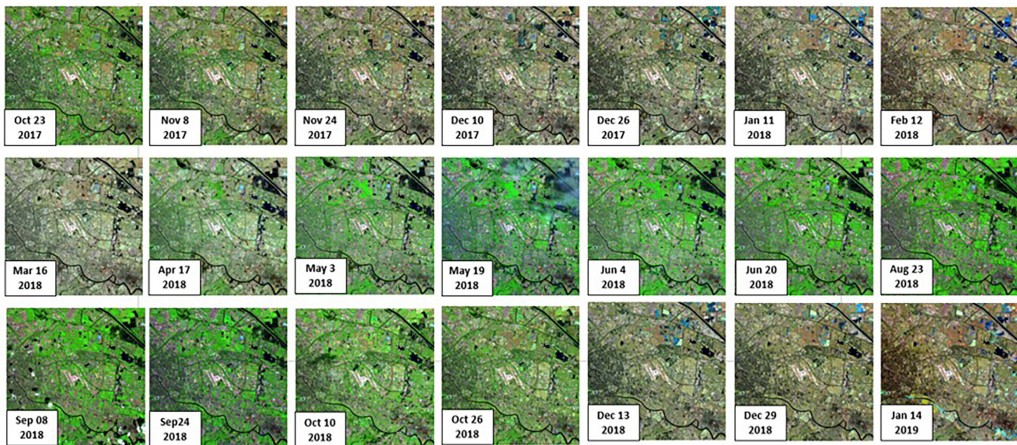

**Figure 4.** False-color composite image of SITS in the study area (R: B7, G: B5, B: B4).

The Pearson correlation coefficient matrix of Landsat-8 SITS in the study area (Tianjin, China) from October 2017 to January 2019 is shown in Tables 1 and 2. The larger the Pearson Correlation Coefficient between different variables, the stronger the linear correlation of these variables, and vice versa. Table 1 shows that the range of Pearson Correlation Coefficient between images both acquired from October 2017 to December 2017 ranges from 0.71~1.00, and the average value is 0.88, indicating that a strong positive linear correlation exists between images with close acquisition times. Table 2 shows that the range of Pearson Correlation Coefficient between images acquired from October 2017 to December 2017 and images acquired from October 2018 to December 2018 is 0.61~0.75, and the average value is 0.67, indicating that the linear correlation between images which acquisition time is far apart is weak. Due to both linear and nonlinear radiation distortions existing between images in SITS in this area, the use of SITS in this area is more beneficial for testing the effectiveness of NMAG model.

**Table 1.** The Pearson correlation coefficients between images both acquired from October 2017 to December 2017.

|  | 2017-10-23 | 2017-11-08 | 2017-11-24 | 2017-12-10 | 2017-12-26 | mean_pccs |
|---|---|---|---|---|---|---|
| 2017-10-23 | 1.00 | 0.89 | 0.83 | 0.79 | 0.71 | 0.84 |
| 2017-11-08 | 0.89 | 1.00 | 0.91 | 0.88 | 0.81 | 0.90 |
| 2017-11-24 | 0.83 | 0.91 | 1.00 | 0.92 | 0.85 | 0.90 |
| 2017-12-10 | 0.79 | 0.88 | 0.92 | 1.00 | 0.89 | 0.90 |
| 2017-12-26 | 0.71 | 0.81 | 0.85 | 0.89 | 1.00 | 0.85 |
| mean_pccs | 0.84 | 0.90 | 0.90 | 0.90 | 0.85 | 0.88 |

**Table 2.** The Pearson correlation coefficient between images acquired from October 2017 to December 2017 and image acquired from October 2018 to December 2018.

|  | 2017-10-23 | 2017-11-08 | 2017-11-24 | 2017-12-10 | 2017-12-26 | mean_pccs |
|---|---|---|---|---|---|---|
| 2018-09-24 | 0.71 | 0.62 | 0.69 | 0.63 | 0.60 | 0.65 |
| 2018-10-10 | 0.73 | 0.66 | 0.75 | 0.68 | 0.65 | 0.69 |
| 2018-10-26 | 0.72 | 0.64 | 0.74 | 0.69 | 0.65 | 0.69 |
| 2018-12-13 | 0.70 | 0.65 | 0.73 | 0.70 | 0.65 | 0.69 |
| 2018-12-29 | 0.67 | 0.61 | 0.69 | 0.68 | 0.64 | 0.66 |
| mean_pccs | 0.70 | 0.64 | 0.72 | 0.68 | 0.64 | 0.67 |

### 3.2. The Preparation of PIFs

PIFs refer to pixels with constant radiometric value in SITS, which were mainly selected from artificial buildings, roads or bare ground pixels. Using high-quality Landsat-8 images of the study area can avoided the interference of clouds and shadows while

selecting PIFs, which help us to select high-quality PIFs. The SITS used here can be expressed as:

$$X = \{X_1, X_2, X_3, ..., X_n\} \tag{11}$$

The least square method was used to estimate the best-fitting parameters ($k_p$ and $b_p$) of the time-series' DN value of pixel $p$ sorted in ascending order $DN_p^X$. Figure 5a shows the distribution of $k_p$ values in the study area. The slope of fitted line of PIFs ($k_s$) is higher than that of water bodies and lower than that of vegetation [20]. Thus, the time-series' DN values of pixel $p$ sorted in ascending order can be given by Equation (12) and the range of $k_s$ is expressed as:

$$DN_p^X = sort\ \{DN_p^{X_1}, DN_p^{X_2}, DN_p^{X_3}, ..., DN_p^{X_n}\}, p \in P \tag{12}$$

$$k_s \in [k_{Low}, k_{High}],\ s \in S \tag{13}$$

where $P$ represents all pixels in images in this area, $S$ represents selected PIFs, and $DN_p^X$ represents the time-series' DN value of pixel $p$ sorted in ascending order.

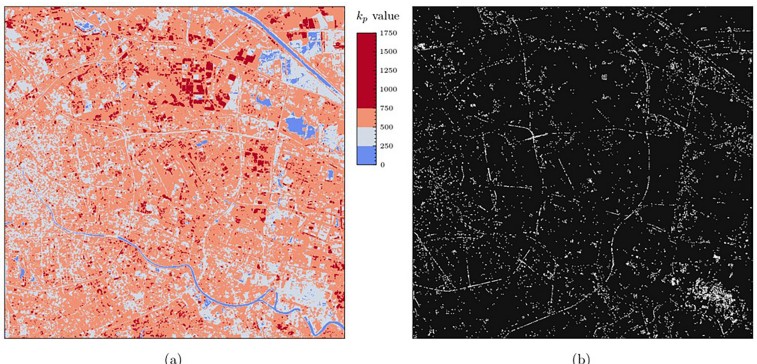

**Figure 5.** The selection result of Pseudo-invariant feature points (PIFs). (**a**) The distribution of $k_p$ values in the study area. The water bodies pixels in light blue; the road, bare land, and urban pixels in gray; and the vegetation pixels are in orange or dark brown. (**b**) The selection result of PIFs. The white pixels represent PIF that is mainly selected from artificial buildings and bare land.

As seen in Figure 5a, water bodies are displayed in light blue, artificial buildings and bare lands are shown in gray, and vegetation is expressed in orange or dark brown. The results indicated that the $k_p$ value of the water bodies is the lowest ($k_p < 250$), the $k_p$ value of the vegetation is the highest ($k_p > 500$), and the $k_p$ values of the artifical buildings and bare lands are between those of the water bodies and the vegetation ($250 < k_p < 500$). PIFs represent pixels with unchanged radiometric values over time, they were mainly selected from artificial buildings and bare lands. Thus, we set the thresholds of $k_s$ used in this paper to extract PIFs were 275 for $k_{Low}$ and 300 for $k_{High}$. The extraction results of PIFs are shown in Figure 5b, and white pixels represent PIFs, with a total of 12,515 pixels. Of these, 8760 pixels are training samples and 3755 pixels are testing samples.

## 4. Results

### *4.1. Experimental Results*

As an illustrative example, we applied NMAG model to the relative radiometric normalization of Landast-8 SITS. Figure 6 shows the normalized results for SITS.

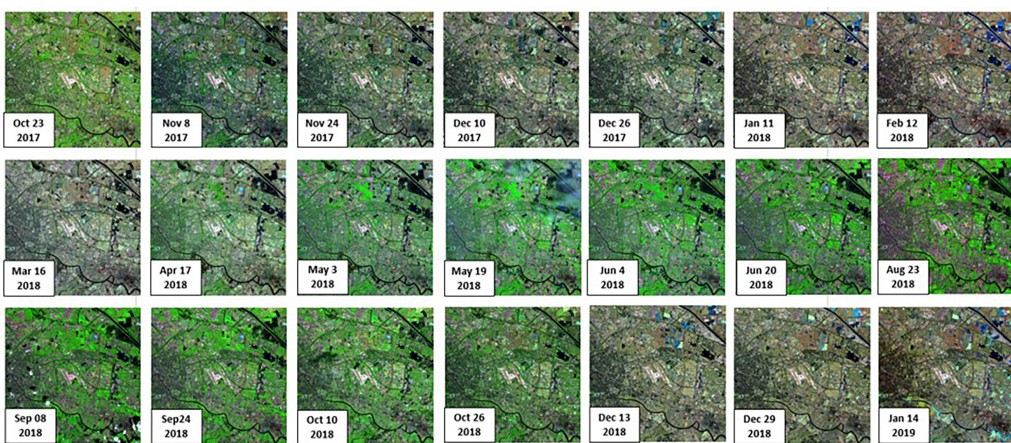

**Figure 6.** The relative radiometric normalized images in SITS.

Figure 7 is a mosaic image composed of four normalized images taken from 10 and 26 December 2017, and 13 and 29 December 2018, respectively. In untreated images shown in Figure 4, there exists significant differences in color intensity and color saturation between images acquired on December 2017 and 2018. However, in normalized images shown in Figure 7a, we can see that all have similar radiometric intensity, hue and color saturation to each other.

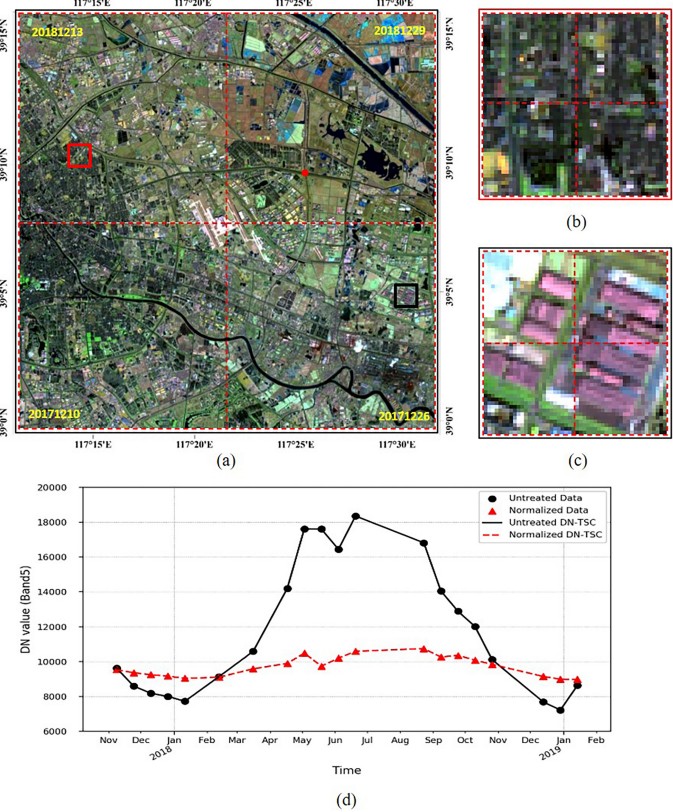

**Figure 7.** (**a**) The mosaic pattern composed of four normalized images taken from from 10 and 26 December 2017, and 13 and 29 December 2018, respectively. The Resultant images have similiar color and color contrast to each other. (**b**)The mosaic pattern of the urban area, and the bundary of this area is marked in red box in **a**. (**c**) the mosaic pattern of the artificial building area, and the bundary of this area is marked in black box in **a**. (**d**) The comparison of time-series' curves of DN values (DN-TSC) from the near-infrared Band (Band 5) for the road pixel (marked in red point in **a**) before (marked in black line) and after (marked in red line) radiometric normalization.

Figure 7b,c are two local area with complex ground features, including artificial bulidings, road and vegetation, are characterized with similiar color and color contrast. These results indicate that the radiometric distortion between different images caused by noise in SITS can be effectively suppressed by NMAG method.

Figure 7d shows the time-series' curve of DN values (DN-TSC) from the near-infrared Band (Band 5) of road pixels before and after radiometric normalization. The untreated DN-TSC (marked by a black line) fluctuates greatly, which indicates that the noise can conceal the actual variation in the spectral characteristics from ground features. After radiometric normalization, the DN-TSC fluctuates slightly and nearly lies to a straight line. This indicates that NMAG effectively suppresses the radiometric distortion between different images in SITS and results in DN values between images in SITS become more comparable.

Figure 8c shows the DN-TSC from the near-infrared Band (Band 5) of pixels in cropland images before (black) and after(red) radiometric normalization. In general, a single peak of DN-TSC exists during the entire corn growth period. According to information obtained from the website of Tianjin Agriculture and Rural Committee (http://nync.tj.gov.cn (accessed on 26 February 2021)) and crop images during the corn growth period (see Figure 8b), we know that corn in this area are generally sown in early June and harvested at the end of October. This means that DN values of cropland pixel is similar to that of bare land pixel from November of the previous year to June of the current year, which should theoretically maintain a fixed value.

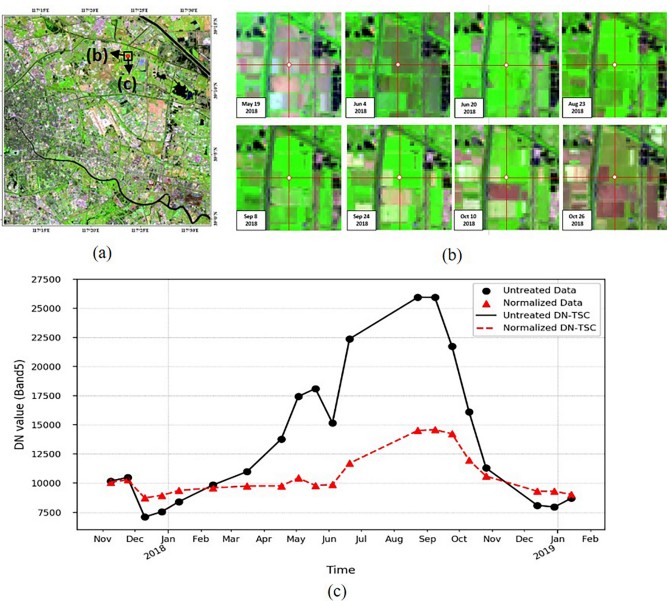

**Figure 8.** The relative radiometric normalization results of DN values of cropland pixels. (**a**) The location of cropland pixels and the boundary of crop images are marked in red point and black box, respectively. (**b**) The cropland images during the growth period of corn. (**c**) The Comparison of the DN-TSC from the near-infrared Band (band 5) for the cropland pixel before (marked by a black line) and after (marked by a red line) the radiometric normalization.

However, the original DN-TSC (marked by a black line) fluctuate greatly from November 2017 to June 2018, and two peaks exists (19 May 2018 and 8 September 2018 ) during the entire corn growth period. This means that the noises contained in the original DN-TSC led to large fluctuations and increased the difficulty of discovering the variation pattern of corn growth.

After radiometric normalization, the DN-TSC (marked in red line) fluctuated slightly from November 2017 to June 2018, demonstrating that NMAG method effectively minimizes the disturbance of noise. A sigle peak (8 September 2018) existed in the normalized

DN-TSC during the entire growth period, which indicated that NMAG can enhance the real time-series' characteristics for cropland pixels.

### 4.2. Comparison with Other Methods

We compared NMAG module with other two contrast methods using the same data. Contrast Method 1 (CM1) proposed by Sadeghi et al. is a nonlinear radiometric normalization method based on the reference image to normalize the subject image [17]. When CM1 is used for the relative normalization of SITS, one image in SITS should be selected as a reference image first and then the other images are normalized to the reference image one by one. The Contrast Method 2 (CM2) proposed by Wu et al. is a radiometric normalization method for SITS based on linearity assumption [20]. This method takes all the corrected images as reference images, and then normalize the subject image by using the Least Squares Method (LSM). To evaluate the accuracy of radiometric normalization results obtained using different methods for SITS, we computed the root mean square error $RMSE(f(X_i), f(X_j))$ to measure the radiometric distortion between the normalized image $f(X_i)$ and the normalized image $f(X_j)$, which is can be calculated from Equation (14). The smaller the $RMSE$ value, the more accurate the radiometric normalization results of images.

$$RMSE(f(X_i), f(X_j)) = \sqrt{\frac{1}{N} \times \sum_{s \in S_t} [f(X_i^s) - f(X_j^s)]^2} \qquad (14)$$

where $X_i$ represents $i$-th image in SITS, $X_j$ represents $j$-th image in SITS. $S_t$ represents the test samples of PIFs, $N$ represents the number of PIF in $S_t$ , $s$ represents the PIF in $S_t$, and $f(X_i^s)$ represents the DN value of PIF in the radiometric normalization result for image $X_i$.

Figure 9a–c shows error matrices of RMSE for the normalized results obtained using NMAG model and two contrasting methods with the same SITS data. In this figure, as the color transitions from blue to red, the RMSE increases. Compared with the error matrices of RMSE obtained using NMAG (Figure 9c), more red pixels are observed in the error matrices of RMSE obtained using CM1 (Figure 9a) and CM2 (Figure 9b). This qualitative analysis result shows that the error matrices of RMSE obtained using NMAG is generally distributed at low values.

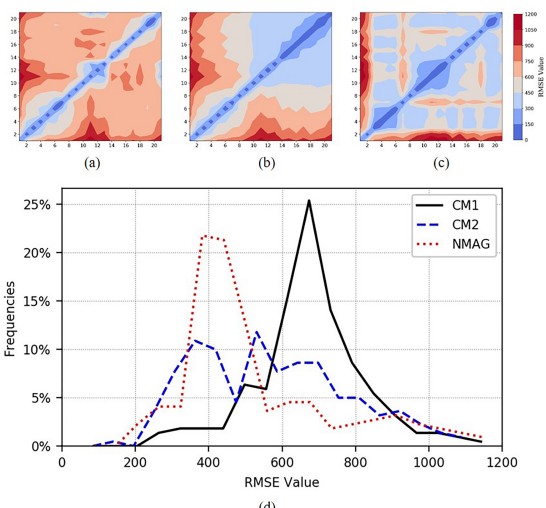

**Figure 9.** The error matrices and the frequency distribution curve of the root mean square error (RMSE) for the normalized results obtained using NMAG and two contrasting methods (refered as Contrast Method 1 (CM1) and Contrast Method 2 (CM2)). As the color transitions from blue to red, the radiometric distortion increases. The error matrices of RMSE obtained using (**a**) CM1, (**b**) CM2, and (**c**) NMAG. (**d**) The frequency distribution curve of RMSE obtained using NMAG and two constrasting methods.

Figure 9d shows the RMSE frequency distribution curve of NMAG (marked by a red line) and other two methods (marked by black line and blue line, respectively). The RMSE between various images of CM1, CM2 and NMAG is concentrated around 700, 550, and 400, respectively. Thus, the error of our method between images in SITS is significantly lower than that of CM1 and CM2. The comparison result demonstrates that NMAG method can reduce the radiometric distortion caused by noise between images in SITS more effectively than two contrasting methods, and obtains more accurate normalized results.

We further compared NMAG method with CM1 and CM2 by calculating the average $RMSE(f(X_i), f(X_j))$. This parameter can be calculated by:

$$\mu_{RMSE} = \frac{1}{n^2} \sum_{i=1}^{n} \sum_{j=1}^{n} RMSE(f(X_i), f(X_j)) \tag{15}$$

where $n$ represents the number of images in SITS. $f(X_i)$ and $f(X_j)$ are respectively $i$-th and $j$-th image in normalized SITS.

Table 3 shows that the $\mu_{RMSE}$ of NMAG (497.22) is significantly smaller than those of CM1 (641.39) and CM2 (543.47), and the accuracy of SITS obtained by using NMAG has increased by 22.4% and 8.5% toward CM1 and CM2, respectively. These results indicate that NMAG model can obtain more accurate SITS.

**Table 3.** Accuracy comparison of the $\mu_{RMSE}$ of NMAG, Contrast Method 1 (CM1) and Contrast Method 2 (CM2).

| Method | CM1 [1] | CM2 [2] | NMAG |
|:---:|:---:|:---:|:---:|
| $\mu_{RMSE}$ | 641.39 | 543.47 | 497.22 |

[1] CM1 proposed by Sadeghi et al. is a nonlinear radiometric normalization method based on the reference image to normalize the subject image [17]; [2] The CM2 proposed by Wu et al. is a radiometric normalization method for SITS based on linearity assumption [20].

### 4.3. Application of NMAG to Vegetation Index

Vegetaion index (VI) obtained from multi-band image data can better reflect the green vegetation status than DN value from single band data. Therefore, the time-series' vegetation index is often used in the field of land cover change monitoring, environmental monitoring and so on [27,28]. As the normalized difference vegetation index (NDVI) is the most frequently used VI in remote sensing applications [29], we further examined the application of NMAG to NDVI. The NDVI can be expressed as:

$$NDVI = \frac{\rho_{NIR} - \rho_R}{\rho_{NIR} + \rho_R} \tag{16}$$

where $\rho_{NIR}$ is the top-of-atmosphere (TOA) reflectance from the near-infrared Band and $\rho_R$ is the top-of-atmosphere (TOA) reflectance from the red Band. With the help of metadata file (_MTL.txt), we can easily transform DN value into TOA reflectance. We used NMAG model to normalize the near-infrared band and the red band separately, and normalized NDVI could be calculated by using normalized result of TOA reflectance of near-infrared Band and red Band.

Figure 10 shows the comparison of time-series' curve of NDVI values(NDVI-TSC) calculated from pixels in corpland images before (black) and after (red) the radiometric normalization. These two NDVI-TSC exist significant differences in details though they have the same trends. The fluctuation amplitude of NDVI-TSC obtained using NMAG is lower than that of untreated curve from November 2017 to May 2018. As mentioned in Section 4.1, the DN values of cropland pixel are similar to those of bare land pixels from November of the previous year to June of the current year, which should maintain a fixed value, theoretically. This means the NDVI value of pixels should be also theoretically maintain a fixed value from November 2017 to May 2018. The reduction in the fluctuation

amplitude of NDVI-TSC indicates that NMAG can effectively decrease noises contained in different bands of SITS.

The times corresponding to the turning point of NDVI-TSC before and after the radiometric normalization are 3 and 19 May 2018. This means that the corn sowing time corresponding to the untreated NDVI-TSC is between 3 and 19 May 2018, and the corn sowing time corresponding to the normalized NDVI-TSC using NMAG is between 19 May and 4 June 2018. In reality, the corn sowing occurred in early June, 2018 according to the information obtained from the website of Tianjin Agriculture and Rural Committee. The normalized result by using NMAG fully matchs the actual situation, showing that the timer-series' NDVI values obtained by using NMAG to normalize SITS data more accurately reflect the vegetation coverage, which can improve the accuracy of remote sensing applications.

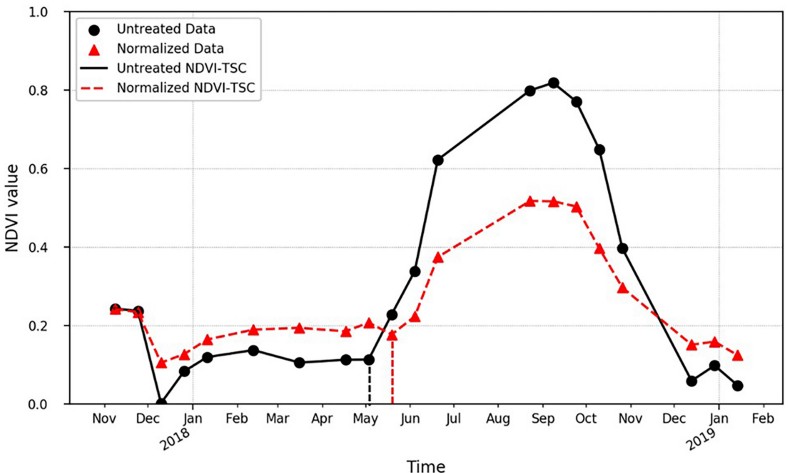

**Figure 10.** The Comparison of time-series'curve of NDVI values (NDVI-TSC) calculated from pixels in corpland images before (marked in black line) and after (marked in red line) the radiometric normalization.

## 5. Discussion

The objective of this study was to supply a model to reduce the nonlinear radiometric distortion caused by continuously changing noise between images in SITS, thereby obtaining a more accurate SITS. So we proposed a nonlinear radiometric normalization method (named NMAG) for SITS based on GA and ANN. A SITS composed of 21 Landsat-8 images in Tianjin, China from October 2017 to January 2019 was used to test NMAG. The experimental results confirmed the effectiveness of NMAG in the reduction of radiometric distortion caused by continuously changing noise between images in SITS. In addition, the performance of NMAG surpasses that of the existing methods (CM1 and CM2) upon comparison.

In theory, NMAG is also suitable for SITS acquired by other satellites, provided that enough relatively invariant pixels can be selected as training samples in the study area. If the number of training samples is insufficient, the model training of NMAG may be insufficient, and the performance of NMAG would be negatively impacted. We recommend that the total number of selected PIFs exceeds 10,000, and the proportion of training samples should not be less than 70%. In the same research area, the higher the spatial resolution of satellite images is, the more convenient the selection of PIFs would be. This means that NMAG may be more convenient for the relative radiometric normalization of satellite image data with higher spatial resolution than Landsat-8. Notably, with increasing number number of PIFs, NMAG model training will take longer, and the performance of NMAG will be increase. Please choose an appropriate number of PIFs according to the hardware performance and the actual needs of the research.

However, NMAG has a certain limitation that may hinder its application. As mentioned above, NMAG needs enough PIFs in the study area to train the model. Thus, NMAG

has weak application potential in study areas with high vegetation coverage or water coverage. Additional studies are needed to find effective models to solve the problem of the scarcity of PIFs that may occur in practical applications.

## 6. Conclusions

In this paper, we propose a nonlinear radiometric normalization method NMAG for SITS based on GA and ANN. This method can effectively suppress the noise contained in SITS, resulting in the gray values of images acquired at different times being more comparable. In the method, GA was used to determine the correction order of SITS and calculate the error between the image to be corrected and corrected images, which avoided the selection of a single reference image. ANN was used to obtain the optimal solution of error function, which minimized the radiometric distortion between images in SITS.

SITS composed of 21 Landsat-8 images in Tianjin, China from October 2017 to January 2019 were used to test our method. The resultant images have similar color and color contrast to each other, and the normalized DN-TSC of the near-infrared Band (B5) obtain by using NMAG can better reflect the actual change of different features than the original DN-TSC. In addition, we compared NMAG with other two existing methods (CM1 and CM2) using the same data. The result shows that the $\mu_{RMSE}$ of NMAG (497.22) is significantly smaller than those of CM1 (641.39) and CM2 (543.47), and the accuracy of SITS obtained by using NMAG has increased by 22.4% and 8.5% toward CM1 and CM2, respectively. This indicates that NMAG can obtain more accurate SITS than other two contrasting methods.

Because the NDVI obtained from multi-band image data can more accurately reflect the green vegetation status than DN value from single band data, we further analyzed the application of NMAG to Vegetation Index. The NDVI-TSC obtained using NMAG to normalize SITS data is fully matched the actual situation, indicating that NMAG can effectively reduce the rediometric distortion caused by noise in SITS so that we can obtain more precise time-series' NDVI values.

**Author Contributions:** All the authors made significant contributions to the work. L.Z., X.S. and Z.Y. designed this research; Z.Y., H.Z. and J.S. implemented the NMAG method and analyzed the results; W.Z. and J.S. designed and performed the comparison experiments between NMAG and other two contrasting methods; Z.Y. wrote the paper; X.S. and L.Z. gave insightful suggestions to the work. All authors have read and agreed to the published version of the manuscript.

**Funding:** This research was founded by the NATIONAL NATURAL SCIENCE FOUNDATION OF CHINA grant No. 41872214 and 42072232.

**Acknowledgments:** The authors would like to express their gratitude to NASA and the USGS for providing remote sensing imageries (Landat-8 OLI images).

**Conflicts of Interest:** The authors declare no conflict of interest.

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
