# Peer review of "A Nonlinear Radiometric Normalization Model for Satellite Imgaes Time Series Based on Artificial Neural Networks and Greedy Algroithm"

_remotesensing, doi:10.3390/rs13050933_

Round 1

Reviewer 1 Report

It is appreciated that the manuscript is easy to follow and not too long. The message is clear and of interest to the community. The authors proposed a nonlinear radiometric normalization method for Satellite Image Time Series data using greedy algorithm (GA) and artificial neural networks (ANN). The performance of their method surpasses the existing methods (CM1 and CM2) upon comparison.

Proposed method seemed to be promising. However, I would suggest a revision before the manuscript could be accepted. Please allow me to clarify.

  1. Please include most recent literature in the introduction section regarding radiometric normalization of satellite time series data.
  2. Please spell out all the abbreviations when first appear, for example CM1, CM2, PIF etc.
  3. In Figure 1 under the processing section, replace refer with reference.
  4. Line 119 – Provide information on how many samples were utilized to train and test the ANN model.
  5. Please tabulate ANN parameter information, such as how many hidden layers, how many neurons in hidden layers etc. This information along with the number of samples being used is going to be critical in accessing the feasibility of ANN modeling in this application.
  6. Please include ANN error convergence graphs for training and validation.
  7. Please include citations in your claim that ANN and GA are the well tested and efficient methods, provided there are other contemporary machine learning models available to handle nonlinear regression modeling using less parameters compared to ANNs, such as SVM and random forest etc.
  8. Line 315 - Please correct the spelling of the word Normalization.
  9. Could you also include in your discussion section about how this methodology could be extended for other multispectral satellite image data such as Sentinel-2, Planet etc.? What are the expected challenges and advantages do you expect, provided their higher spatial resolution compared to Landsat-8?

Reviewer 2 Report

In this paper, a nonlinear radiometric normalization model for SITS based on Artificial neural networks and a greedy algorithm is proposed.  In general terms, this work is very interesting, it is well written, structured and it is clear about the significance and its novelty; however, I consider that the manuscript presents some drawbacks that need to be addressed before its publication in this journal. Namely,

  1.    The greedy algorithm used in this paper should be included and briefly explained in the manuscript. 
    2.    The technical specifications of the neural network employed in this work should be detailed, in order to give to potential readers more information about how the proposed method work.
    3.    It is necessary to summarize the main differences of the proposed method and the one presented in reference [14]. 
    4.    Finally, the experimental section should be improved, in order to give enough details for its possible replication.

Minor issues 

Check lines 82, 85, 100 (Eq. 5), 183, 215, and reference style.

Round 2

Reviewer 1 Report

Authors have done a fine job responding to my comments and incorporating my suggestions. I highly recommend the manuscript for publication.

nicepleasant or pleasing or agreeable in nature or appearanceMore (Definitions, Synonyms, Translation)